# Diagnostic accuracy of three commercially available one step RT-PCR assays for the detection of SARS-CoV-2 in resource limited settings

**Abay Sisay** [1,2☯]*, **Adugna Abera** [2,3☯], **Boja Dufera** [2,3], **Tujuba Endrias** [3],
**Geremew Tasew** [3], **Abraham Tesfaye** [1,4], **Sonja Hartnack** [5], **Dereje Beyene** [2], **Adey Feleke Desta** [2]

1 Department of Medical Laboratory Sciences, College of Health Sciences, Addis Ababa University, Addis Ababa, Ethiopia, 2 Department of Microbial, Cellular and Molecular Biology, College of Natural and Computational Sciences, Addis Ababa University, Addis Ababa, Ethiopia, 3 Malaria and Neglected Tropical Diseases Research Team, Ethiopian Public Health Institute, Addis Ababa, Ethiopia, 4 Diagnostic Unit, Center for Innovative Drug Development and Therapeutic Trials for Africa, CDT- Africa, Addis Ababa, Ethiopia, 5 Section of Epidemiology, University of Zurich, Zurich, Switzerland

☯ These authors contributed equally to this work.
* abusis27@gmail.com

**Data Availability Statement:** All relevant data are within the manuscript and its Supporting Information files

## Abstract

### Background

COVID-19 is an ongoing public health pandemic regardless of the countless efforts made by various actors. Quality diagnostic tests are important for early detection and control. Notably, several commercially available one step RT-PCR based assays have been recommended by the WHO. Yet, their analytic and diagnostic performances have not been well documented in resource-limited settings. Hence, this study aimed to evaluate the diagnostic sensitivities and specificities of three commercially available one step reverse transcriptase-polymerase chain reaction (RT-PCR) assays in Ethiopia in clinical setting.

### Methods

A cross-sectional study was conducted from April to June, 2021 on 279 respiratory swabs originating from community surveillance, contact cases and suspect cases. RNA was extracted using manual extraction method. Master-mix preparation, amplification and result interpretation was done as per the respective manufacturer. Agreements between RT-PCRs were analyzed using kappa values. Bayesian latent class models (BLCM) were fitted to obtain reliable estimates of diagnostic sensitivities, specificities of the three assays and prevalence in the absence of a true gold standard.

### Results

Among the 279 respiratory samples, 50(18%), 59(21.2%), and 69(24.7%) were tested positive by TIB, Da An, and BGI assays, respectively. Moderate to substantial level of

**Funding:** This work was financed by Addis Ababa University through adaptive research and problem solving project, Ref #-PR/5.15/590/12/20. However, the funders had no role in study design, data collection and analysis, decision to publish, or preparation of the manuscript. The contents are purely the responsibilities of the authors and did not represent and reflect the view of the funder. There was no additional external funding received for this study.

**Competing interests:** The authors have declared that no competing interests exist.

**Abbreviations:** COVID-19, Corona virus disease; Ct-value, Cycle threshold; Da An, Da An Gene Co., Ltd. of Sun Yat-sen University; DNA, Deoxyribonucleic acid; NNAF, Nucleic Acid Amplification Test; ORF, Open reading frame; PCR, Polymerase chain reaction; RdRp, RNA-dependent RNA polymerase; RNA, Ribonucleic Acid; RT-PCR, Reverse transcriptase polymerase chain reaction; SARS-CoV-2, Sever acute respiratory syndrome corona virus; VTM, Viral transport Medium; WHO, World Health Organization.

agreement was reported among the three assays with kappa value between 0 .55 and 0.72. Based on the BLCM relatively high specificities (95% CI) of 0.991(0.973–1.000), 0.961 (0.930–0.991) and 0.916(0.875–0.952) and considerably lower sensitivities with 0.813 (0.658–0.938), 0.836(0.712–0.940) and 0.810(0.687–0.920) for TIB MOLBIOL, Da An and BGI respectively were found.

## Conclusions

While all the three RT-PCR assays displayed comparable sensitivities, the specificities of TIB MOLBIOL and Da An were considerably higher than BGI. These results help adjust the apparent prevalence determined by the three RT-PCRs and thus support public health decisions in resource limited settings and consider alternatives as per their prioritization matrix.

## Introduction

Coronavirus disease 2019 (COVID-19) is still an ongoing global pandemic and a public health emergency resulting in more than 4,891,942 deaths from 240,061,637 documented cases as of 13 October, 2021 the first reported case in China, Wuhan in December 2019 [1,2].

The global community had three epidemic surges of Severe Acute Respiratory Syndrome Coronavirus 2 (SARS-CoV-2) causing COVID-19. This has imposed an immediate increasing demand on daily diagnostic screening, with the aim to shorten the epidemic curve and to prevent the occurrence of new variants of concern and interests, which could lead the next wave as has, occurred in some parts of the globe [3,4]. It is a pushing factor and time for designing and a critical need for accurate and rapid diagnostic assays to prompt clinical and public health interventions. This can help to perpetuate over the upcoming time, due to the recurrence of outbreaks [5]. One step reverse transcriptase PCR is a popular diagnostic technique for novel Corona virus ever since the first case of COVID-19 reported elsewhere [6,7].

In Africa, including Ethiopia, COVID-19 laboratory testing facilities are centralized, that have been managed in a referral linkage of a limited access to laboratory facilities, by which there is a possibility of overloading and flooding of the already overwhelmed reference laboratories during emergency conditions. Availability, low cost and user friendliness are highly desirable in Ethiopia. While high sensitivity is of paramount importance not to miss any COVID-19 case and stop virus transmission [8], diagnostic specificity is highly relevant when tests are used as confirmatory and to avoid the costs of false positive results. In most COVID-19 diagnostic tests accuracy studies, RT-PCR is considered as the gold standard, i.e. having 100% diagnostic sensitivity and specificity. Despite the tremendous worldwide research activities related to COVID-19 diagnostic tests, very few diagnostic test accuracy studies were conducted to adjust imperfect tests [9–13]. So far, very few attempts were made to assess the diagnostic performance of RT-PCRs in Ethiopia in a clinical setting.

### COVID-19 pandemic in Ethiopia

Ethiopia is the second most populous country in Africa. It is one of the 220 COVID-19 affected countries and territories, since the first confirmed COVID-19 was reported on March 13, 2020 [14]. Presently case numbers are increasing rapidly. There were two peaks of kurtosis of the epidemic; first wave from August to September 2020, and second wave occurred during February to March 2021. There are reports of Delta variants in parts of Africa including Ethiopia

which might have resulted in the third wave [15,16]. As of 13 October 2021 there are **356,772** confirmed cases which make Ethiopia the 4[th] most affected country in Africa [15] with **6,103** documented deaths. A total of **3329** health professionals have contracted the disease and 36 of them have died [15,17].

PCR-based laboratory testing is indispensable in identifying the infected cases and also curbs the transmission epidemic. In Ethiopia, so far there are **3,574,218** total tests using RT-PCR, which equates to **30,136** tests per million population, since the first COVID-19 case was reported and the subsequent declaration of global pandemic [15,17]. Even though the country has come a long way in increasing the testing capacity through redirecting and mobilizing resources and organizing testing campaigns, the tests per the total population are still very low and need due attention [18,19].

## Available diagnostic platforms for the diagnosis of COVID-19

Since the occurrence of the current pandemic: Severe Acute Respiratory Syndrome Coronavirus-2 (SARS-CoV-2) infection on December, 2019 in Wuhan, China, Scientist have been developing diagnostic tools with different diagnostic tests, relying on different biological principles [20].

Detection methods based on nucleic acid amplification tests (NAAT) are popular and preferable for novel Coronavirus in many countries. The test demonstrated highest sensitivity when it performed at the earliest time point in the acute phase of infection. The current reference or gold standard for diagnosing COVID-19 depends up on detection of the viral genetic material (RNA) in a nasopharyngeal swab or sputum sample., it requires PCR, a technology that amplifies the amount of genetic material to detectable levels and takes several hours to perform [21].

One-step reverse transcriptase-polymerase chain reaction (RT-PCR) is the established reference standard test for the diagnosis of COVID-19 because of its performance in identifying SARS-CoV-2 infection. This technique is sensitive and can detect the virus earlier in the infection [21]. Currently, there are different RT-PCR assays with quite different analytical and diagnostic sensitivity and specificity have been developed and recommended by the WHO and used for the diagnoses of SARS-CoV-2 [22,23].

Evidence indicated that, in many countries all available test kits have been used including rapid diagnostic tests that detects the virus either an antibody or an antigen based, whereas Ethiopia uses both RT-PCR and Ag RDT for the diagnosis of SARS CoV- 2 [8,18,24]. Indeed, evidence based information on the diagnostic performance of the commercially available test assays in the country and currently used by most COVID-19 testing laboratories are critical and timely for the program managers and have its impact in the containment strategy of the virus through quality test kit selection on their prioritization and continuous quality improvements on its development as part of the control of the pandemic [9].

In view of this, many pharmaceutical companies have accomplished different activities across the world as part of the pandemic response, which can benefit synergetic modality. However, so far because of its analytical and diagnostic presentation, RT-PCR is the reference method for the detection of SARS-CoV-2, irrespective of their variability in performance [25]. As of September 11, 2021 there are 1129 SARS-CoV-2 tests that are commercially available or in development for the diagnosis of COVID-19 and evaluated through FIND in collaboration with WHO and partners. FIND has been conducting independent evaluation of the available assays and summarized data for emergency use authorization (EUA) [23].

Testing for SARS–CoV-2 is becoming increasingly available in high-income countries and though still scant in resource limited countries, and even when available, it is highly centralized and limited to more specialized institutions with greater supply chain bottlenecks [26].

Reverse transcriptase-PCR test is usually done to identify positive cases in a population to prevent transmission to others and monitoring patient outcome. Likewise, it is critical importance for the mitigation of the pandemic. This is very much pondering if and only if the laboratory used better comparable quality diagnostic assays [27–30].

In Ethiopia, there are considerable numbers of one step RT-PCR assays commercially available for the diagnosis of COVID-19 either by donation or direct procurement by government. Among the available assays, three of them Sarbeco virus SARS-CoV-2 TIB MOLBIOL (hereafter named as TIB), Da An and BGI are widely used for the diagnosis of SARS CoV-2 at different time. TIB assay has two targets, E and RdRp involving two steps. The first step is screening by the E gene and the second step is confirming all positive cases by the RdRp gene. Da An assay works by multiplexing and targets two viral genes (nucleocapsid (N) and the ORF 1ab). The BGI assay targets only one viral gene the ORF 1ab. Each assay has its own procedural and result interpretation steps. However, their performance in a clinical setting in Ethiopia has not been assessed.

As PCR is not considered as gold standard our main scientific hypothesis is the assessment of comparing the ct levels and then the demographic data, and finally assessing agreement between the PCRs and obtaining robust estimates for diagnostic sensitivities and specificities in the less developed countries.Therefore, the current study was aimed to evaluate the diagnostic sensitivities and specificities of three commercially available one step reverse transcriptase assays for the detection of SARS-CoV-2 in resource limited settings, Ethiopia using Bayesian latent class model (BLCM).

## Materials and methods

### Study design, period and settings

A health facility based cross sectional study was conducted at Parasitology COVID-19 testing laboratory of Ethiopian Public Health Institute (EPHI) from April to June 2021. Nasopharyngeal swabs were collected from 279 individuals referred for COVID-19 testing from health facilities of Addis Ababa. The laboratory is amongst the first established national COVID-19 testing laboratory with qualified and competent performance which has been serving COVID-19 testing for samples come from referring health facilities of Addis Ababa through the national emergency operation center (EOC) [18].

**Flow of study participants.** In this study a total of 468 eligible participants who fulfill the WHO criteria for COVID-19 suspected cases were screened [2]. Of these a total of 279 nasopharyngeal swabs collected by trained health professionals who encompass from community surveillance consider them as population one (32.62%), contacts of confirmed cases, consider as population two) (17.2%), and suspects, consider them as population 3 (50.18%). The detail is depicted at diagram 1 in S1 Annex.

**Test kits selection.** Reverse transcriptase-PCR test has been designed based by targeting some parts of SARS-COV-2: spike (S), open reading frame1a/b (ORF1a/b), ORF1b-nuclear shuttle protein14 (ORF1b–nsp14)), envelope (E), RNA-dependent RNA polymerase (RdRp) or nucleocapsid (N) genes from the structural and nonstructural part of the gene [27]. Accordingly, the current study evaluated test kits employed E gene, RdRp, N gene and ORF as illustrated in Table 1. However, their respective pharmaceutical companies had not had any involvement with the research methodology design, analysis and write up of the research manuscript.

**Laboratory procedures.** *RNA purification*. Respiratory specimens (throat/nasopharyngeal) collected in 2 mL VTM (China, Miraclean Technology Co.,Ltd., www.mantacc.com) by well trained professionals were used for RNA purification. Briefly, 200 μL of respiratory samples were transferred into 1.5 ml eppendorf tube. Then, 50 μl of proteinase K and 200 μl of lysis buffer were added after brief centrifugation and incubation of the tube at 72˚C for 10

**Table 1. RT-PCR test kits with their corresponding primers and probes sequences used in this study, Addis Ababa, Ethiopia, 2021.**

| S# | Assay name and company | # of targets | Target Gene | Primers and probes sequence information (5'-3') | Length (bases) | Ref |
|---|---|---|---|---|---|---|
| 1 | TIB MOLBIOL, Germany * | 2 | E gene (Screening) | Forward primer: ACAGGTACGTTAATAGTTAATAGCGT<br>Reverse primer: ATATTGCAGCAGTACGCACACA<br>Probe: FAM ACACTAGCCATCCTTACTGCGCTTCG-BBQ | 145 bp | [31,32] |
| | | | RdRp gene (Confirmatory) | Forward primer: ATGAGCTTAGTCCTGTTG<br>Reverse primer: CTCCCTTTGTTGTGTTGT<br>Probe: FAM-AGATGTCTTGTGCTGCCGGTA [5']Hex [3'] BHQ-1 | 196 bp | [32] |
| 2 | Da An Gene Co., Ltd. of Sun Yat-sen University, China | 2 | N gen | Forward primer: TAATCAGACAAGGAACTGATTA<br>Reverse primer: CGAAGGTGTGACTTCCATG<br>Probe: FAM/ZEN-GCAAATTGTGCAATTTGCGG-IBFQ | 323 bp | [32,33] |
| | | | ORF1ab | Forward: AGAAGATTGGTTAGATGATGATAGT<br>Reverse: TTCCATCTCTAATTGAGGTTGAACC<br>Probe:FAM TCCTCACTGCCGTCTTGTTGACCA-BHQ1 | 588 bp | [32,34] |
| 3 | BGI | 1 | ORF1ab | Forward primer: AGAAGATTGGTTAGATGATGATAGT<br>Reverse primer: TTCCATCTCTAATTGAGGTTGAACC<br>Probe:FAM-TCCTCACTGCCGTCTTGTTGACCA-BHQ1 | 588 bp | [32,34] |

**Note:-** FAM: 6-carboxyfluorescein; BBQ: blackberry quencher; BHQ-1: Black Hole Quencher-1;. W is A/T; R is G/A; M is A/C; S is G/C, RdRp: RNA-dependent RNA polymerase

*we have used WHO TIB reagent. First line screening assay: E gene assay, Confirmatory assay: RdRp gene assay.

minutes. 250 μl of absolute alcohol was added and the whole mixture was transferred to spine column. The remaining procedures were performed as per the Da An RNA/DNA purification kit insert (Da An Gene Co., Ltd. of Sun Yat-sen University). In all extraction procedures, positive and negative controls were included.

**Detection of SARS-CoV-2 by RT-PCR.** Following the Viral RNA extraction the extracted samples were detected by the three (TIB, Da An and BGI) commercially available RT-PCR assays against their respective manufacturer recommendations and instructions on a Quant Studio 5 DX real-time PCR system (catalog number A34322, Thermo Fisher Scientific), and data were analyzed and computed. Master Mix preparation and cycling conditions for respective assays presented in Table 2.

**Determination of results.** For the TIB assay, a 76 bp long fragment from the E gene is amplified with specific primers and detected with a FAM label hydrolysis probe. The assay detects SARS and 2019-nCoV pneumonia virus (bat-associated SARS related Sarbeco virus has its own control reaction of 70 bp fragment from Equine Arteritis Virus detected with an Atto647 labeled probe. It's RNA positive control contains all diagnostic targets E gene, N gene and RdRP. Mainly it had FAM Color Module 1 and Cy5: Color module 5 (IC) by RdRp also FAM: Color Module 1 and Cy5: Color module 5 (IC) by E gene. It has two steps to declare positive: screening test done by E gene and if any clinical sample positive by the E-gene assays are subject to repeat and confirmatory tests done by the RdRp gene-specific to SAR-CoV-2 [35].

For Da An assay, if the test sample has no amplification curve in the FAM and VIC channels, but amplification curve in the Cy5 channel, the result can be judged as there is no 2019 Novel Coronavirus (2019-nCoV) RNA in the sample; If the test sample has obvious amplification curve in the FAM and VIC channels and Ct values ≤40, the result can be judged as the sample is positive for 2019 novel coronavirus; if the test sample only has the ct value ≤40 in a single channel of FAM or VIC and there is amplification curve in Cy5 channel, the results need to be re-tested. If the re-test results are consistent, the sample can be judged as positive for 2019 Novel Coronavirus (2019-nCoV). If the re-test results are negative, it can be judged

**Table 2. RT-PCR test kits with their corresponding master mix protocols used in this study, Addis Ababa, Ethiopia, 2021.**

| Types of assay | Master mix | Volume of reagent used per reaction | Cycling conditions | |
|---|---|---|---|---|
| TIB (EAV/RdRp) | Nuclease free water | Nx3.3 μl | Initial RT step 50˚C for 30 minutes | |
| | Combined primer/Probe mix | N x 0.5 μl | 95˚C 2 minutes | |
| | Internal control | N x 0.5 μl | | |
| | qScript One-Step enzyme | N x 0.7 μl | 95˚C for 15 seconds | X45 cycles |
| | 2X one Step master mix | N x 10 μl | 55˚C for 30 seconds | |
| | Total volume | N x 15 μl | | |
| | Template per each reaction tube | 5 μl | | |
| Da An | Solution A | N x 17 μl | Initial RT step 50˚C for 15 minutes | |
| | Solution B | N x 3 μl | 95˚C 15 minutes | |
| | Total volume | N x 20 μl | 95˚C for 15 seconds | X45 cycles |
| | Template per each reaction tube | 5 μl | 55˚C for 30 seconds | |
| BGI | Reaction Mix | N x 18.5 μl | Initial RT step 50˚C for 15 minutes | |
| | Enzyme mix | N x 1.5 μl | 95˚C 15 minutes | |
| | Total volume | N x 20 μl | 95˚C for 15 seconds | X40 cycles |
| | Template per each reaction tube | 10 μl | 55˚C for 30 seconds | |

that no 2019 Novel Coronavirus (2019-nCoV) RNA has been detected. The ROC curve method is used to determine both the reference Ct value of the kit and the internal standard reference values are 40 for positive value judgment [36].

*Interpretation of test results.* in each test procedure, negative, positive and no template controls were included. Only when the controls meet the requirements can the test results be determined. When the FAM and VIC detection channels are positive, the result from the Cy5 channel (internal standard channel) may be negative due to the competition of the system and when the internal standard result is negative, if the FAM and VC detection channels of the test tube are also negative, it indicates that the system is inhibited or the operation is wrong, the test is invalid. Therefore, the sample needs to be re-tested [36].

In case of BGI, the controls for the Real-Time Fluorescent RT-PCR Kit for Detecting SARS-2019-nCoV are evaluated using the nucleic acid amplification curve and Ct values generated by the RT-PCR system. Ct cut-off values were determined using the receiver operator characteristic curves of the tested clinical samples. The Ct value in the FAM channel for a valid no template (negative) control should be "0" and there should be no sigmoidal amplification curve. Experimental analysis found that Ct value for positive SARS-CoV-2 samples should be no higher than 38. Thus, the Ct value in the FAM channel for a valid positive control should be no higher than 38 and there should be a sigmoidal amplification curve. Experimental analysis found that Ct value for the internal positive control samples should be no higher than 35. Thus, the Ct value in the VIC/HEX channel for a valid internal positive control should be no higher than 35 and there should be a sigmoidal amplification curve [37].

**Data quality assurance.** Sample and data was collected by well trained professionals. All laboratory procedures were performed as per the documented standard operating procedures and according to specific manufacturing recommendations. The quality of each reagent like lot number, expire date, storage conditions, physical leak proof and not breakages, etc were cheeked before the actual laboratory analysis. Samples and reagents were stored at appropriate temperature as indicted on the manufacturer inserts. Internal and external quality controls were run as required during laboratory analysis.

Data was double checked manually for completeness and consistency before data entry. We have used a respective recommended manufacturer protocols. The above requirements must

be met at the same time in each experiment; otherwise, the experiment is invalid and needs to be carried out again.

**Data analysis.** For the best narrative, only results findings by three RT-PCR tests were analyzed and considered for our further analysis and interpretation, the rest were trimmed from the final analysis. Results were interpreted as positive and negative based on the cut-off Ct values of each manufacturer. The binomial 95% confidence intervals (95% CI) were calculated using MedCalc (MedCalc Software Ltd.) and P-values <0.05 considered statistical significant using Vassarstats and SPSS v.23 statistical software Cohen's kappa statistic among the three assays. A value of 1 implies almost perfect agreement and values less than 1 imply less than perfect agreement, with a range of values between 0 and 1 [38].

Diagnostic sensitivity, specificity, predictive and kappa values were computed. Positive and negative predictive values were computed following.

**Bayesian latent class model (BLCM).** With the aim to obtain diagnostic tests accuracies in the absence of a perfect gold standard, a Bayesian latent class model (BLCM) was fit to the data following the approach from Hui and Walter for three tests and three populations using MCMC (Markov chain Monte Carlo) simulation to construct posteriors in JAGS version 4.3.0 [39] using the runjags package [40].

The frequencies of the eight combinations of dichotomized RT-PCR results (+++; ++-; +-+;+ —;-++,-+-,—+,—) in the three populations, respectively, were modeled with a multinomial distribution. To allow for potential conditional dependencies, pair wise covariances between sensitivities and specificities of all RT-PCRs were included in separate models. Model selection, i.e. in- or exclusion of conditional dependencies was based on the 95% credibility intervals (including 0 or not) and on Bayesian p-values. The model code (S1 File) was obtained with the function "auto huiwalter" of the runjags package [41], with three chains of 50 000 iterations each, a burn-in of 5000 iterations, and a thinning of 10 iterations. Minimally informative priors (beta (1,1)) were used for the sensitivities and prevalence's. In contrast, for the specificities informative priors (beta (34.17, 1.33)) were included. The shape parameters were obtained with beta buster assuming "to be 95% sure that the specificity is greater than 90% with a mode at 99%". Convergence was assessed by visual inspection of the trace plots and the potential scale reduction factor (Gelman Rubin statistic) being below 1.1. A sensitivity analysis was performed by using different combinations of minimally (dbeta(1,1)) or weakly informative priors (dbeta(2,1)).

**Ethical consideration.** This research has been accomplished after getting ethical approval from EPHI and AAU (EPHI–IRB-259-2020 and 029/20/Lab (AAU, CHS) respectively). All participants' identifiers were removed and only codes were used throughout the study to keep confidentiality.

During data and sample collection the data collectors inform each study participant about the purpose and anticipate benefits of the research project and also informed on their full right to refuse, withdraw or completely reject partly or all of their part in the study. Then, we obtained written informed consent from adult study participants and parents or legal guardians of study participants under the age of 18 years to participate in the study and to use their files and records for the purpose of this study.

Furthermore, this work was done accordance with the principles of Helsinki declaration.

## Results

### Laboratory results of study participants by the three assays

A total of 279 respiratory samples were analyzed using by three assays. The age distribution of the participants ranges from 10 to 85 years with majorities (40%) of the participants were 21–30 years and 144(51.6%) of the participants were female. We have three different populations,

**Table 3. Socio demographic characteristics of study participants against the three assays in Addis Ababa, Ethiopia, 2021(N = 279), 2021.**

| Variables | Assay | | | | | |
|---|---|---|---|---|---|---|
| | TIB | | DaAn | | BGI | |
| | Neg | Pos | Neg | Pos | Neg | Pos |
| **Sex** | | | | | | |
| Female | 115 | 29 | 110 | 34 | 106 | 38 |
| Male | 114 | 21 | 110 | 25 | 104 | 31 |
| **Age in Year** | | | | | | |
| <10 | 6 | 4 | 7 | 3 | 5 | 5 |
| 11–20 | 14 | 13 | 15 | 12 | 13 | 14 |
| 21–30 | 64 | 25 | 69 | 20 | 62 | 27 |
| 31–40 | 51 | 16 | 49 | 18 | 43 | 23 |
| 41–50 | 20 | 12 | 21 | 11 | 18 | 14 |
| 51–60 | 24 | 5 | 22 | 7 | 19 | 10 |
| >60 | 20 | 5 | 18 | 7 | 18 | 7 |
| **Reason for testing** | | | | | | |
| Community surveillance | 89 | 2 | 87 | 4 | 84 | 7 |
| Contact of confirmed case | 35 | 13 | 32 | 16 | 32 | 16 |
| Suspect | 104 | 34 | 100 | 38 | 93 | 45 |

of which half of the participants samples 140(50.18%) were from suspect group followed by community surveillance 91(33%) and the remaining from close contacts, Table 3.

Results with threshold cycle (Ct) values >40 or those that remained undetected during the 45 cycles of the experiment work and a probability of false negatives using these three test assays with a positivity rate of from 18 to 25%. There were 2 more cases were positive by E gene (the screening test of TIB) while become negative by RdRp (the confirmatory one). The results were evaluated in comparison with those of the TIB™, Daan™ COVID-19 Direct assay and the BGI COVID-19 assay as indicated Fig 1 and Table 4.

The performance of each assay was evaluated and Ct values of 2019-nCoV positive samples ranged from 12.4 to 37.6 and 13.3–38.7 for RdRP and E genes of TIB, 13.3–39.22,16.65–38.91 for N genes and ORF of DaAn and from 10.9–35.5 of ORF of BGI. The value for the ORF gene

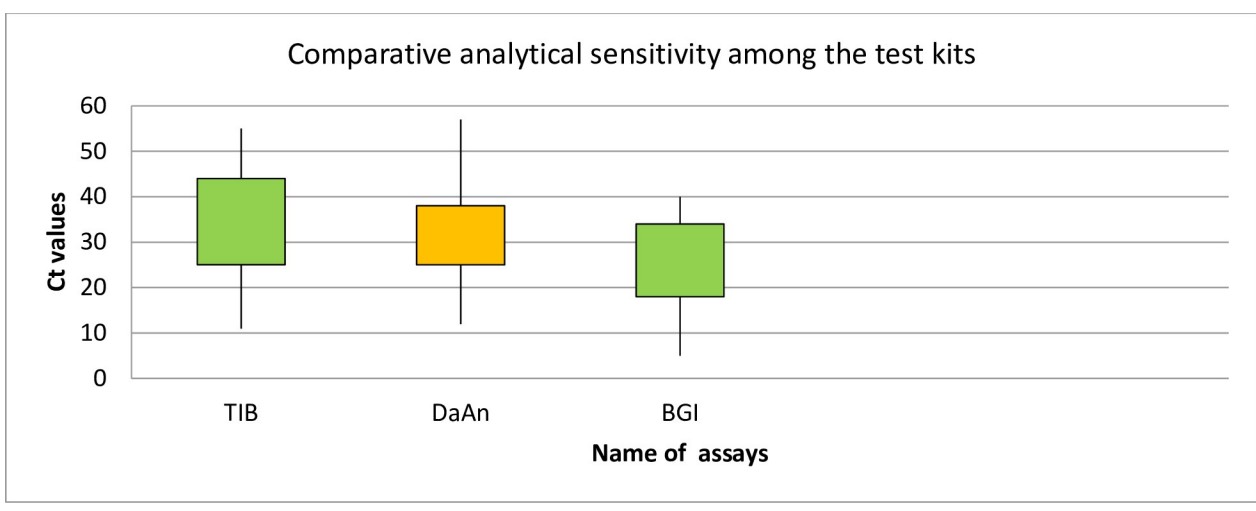

**Fig 1. Comparative analytical sensitivity of the three RT-PCR assays based on Ct values, Addis Ababa, Ethiopia, 2021.**

**Table 4. Overall results of 279 samples tested by three molecular SARS-CoV-2 detection assays value with corresponding Ct values, Addis Ababa, Ethiopia, 2021.**

| Variables | TIB MOLBIOL | DaAn | BGI | mean (± SD) Ct value |
|---|---|---|---|---|
| **Positive results** | | | | |
| All(ct value) | 52 (17.9–39.4) | 48 (17.5–39.2) | 55 (18.7–39.3) | 30.1±5.8 |
| E gen and RdRp | | | | |
| Egenes(ct value) | 52(9–38.7) | | | 28.81±6.4 |
| RdRp(ct value) | 50(12.4–37.6) | | | 29.54±6.1 |
| N gen and ORF 1 | | | | |
| N gen(ct value) | | 59(13.3–39.2) | | 31.95±5.6 |
| ORF1(ct value) | | 60(16.6–39.4) | | 34.20±5.6 |
| ORF(ct value) | | | 69(10.9–35.5) | 28.78±5.7 |
| **Negative result** | 229 | 219 | 210 | |
| No. of concordant results | 50 | 60 | NA | |
| # of discordant results | 2 | 1 | NA | |

NOS nasal/oropharyngeal swab, C threshold cycle, E envelope, RdRP RNA-dependent RNA polymerase, N nucleocapsid, S spike, ORF open reading frame.

of BGI (28.78±5.7), which makes the lowest ct value among the average of the positive samples.

The raw data, comprising the frequencies of the eight possible combinations of dichotomized test results of the three RT-PCRS in the three populations are displayed in Table 5A.

We perform a Cohen's kappa coefficient(k) to see the inter test agreement beyond chance of the three assay using 279 COVID-19 suspected individuals and we have a moderate to substantial level of agreement finding report among the three RT-PCR test kits with a kappa value of 0 .55 to 0.72 as the detail illustrated in Table 5B.

Thus, since none of the currently available COVID-19 tests can be considered as a perfect gold standard with both a perfect sensitivity and specificity, we employed a Bayesian latent class model (BLCM) using Markov Chain Monte Carlo sampling with Just another Gibbs sampler (JAGS) to the clients' data using three tests in three populations.

**Table 5. a. Possible frequencies combination of dichotomized test results of the three assays among three populations in Addis Ababa, Ethiopia, 2021.** b. Measure of Cohen's kappa coefficient (κ) Value among the three tests kits, Addis Ababa, Ethiopia, 2021.

| TIB | Da An | BGI | Population 1 | Population 2 | Population 3 |
|---|---|---|---|---|---|
| - | - | - | 83 | 26 | 81 |
| - | - | + | 4 | 4 | 14 |
| - | + | - | 1 | 2 | 8 |
| - | + | + | 1 | 3 | 2 |
| + | - | - | 0 | 1 | 2 |
| + | - | + | 0 | 1 | 4 |
| + | + | - | 0 | 3 | 3 |
| + | + | + | 2 | 8 | 26 |
| | | | Total: 91 | Total: 48 | Total: 140 |

| Inter rater tests | Cohen's kappa coefficient (κ) Value: | 95% Confidence Interval(CI) | | p-value | Interpretation |
|---|---|---|---|---|---|
| | | Lower Limit | upper Limit | | |
| Measure of Agreement among test 1 and test 2 | 0.72 | 0.61 | 0.82 | <0.001 | substantial agreement |
| Measure of Agreement among test 1 and test 3 | 0.61 | 0.49 | .0.72 | <0.001 | substantial agreement |
| Measure of Agreement among test 2and test 3 | 0.55 | 0.44 | 0.67 | <0.001 | Moderate agreement |

**Table 6. Posteriors means and corresponding 95% credible intervals (CrI) of BLCM models with different pair wise covariance's added, Addis Ababa, Ethiopia,2021.**

|  | Models | | | | | | |
|---|---|---|---|---|---|---|---|
|  | no | Cov.se12 | Cov.se13 | Cov.se23 | Cov.sp12 | Cov.sp13 | Cov.sp23 |
| Posteriors |  |  |  |  |  |  |  |
| Se_TIB | 81.2 [65.8;93.8] | 79.7 [60.1;94.4] | 81.0 [65.4;94.4] | 81.0 [66.2;94.2] | 78.9 [63.0;93.0] | 79.3 [64.0;92.7] | 79.9 [63.6;93.5] |
| Se_Da An | 83.6 [71.2;94.0] | 82.1 [65.0;94.5] | 83.6 [69.9;95.2] | 84.0 [71.1;94.3] | 80.9 [68.3;92.3] | 83.2 [70.8;94.3] | 82.7 [70.3;93.6] |
| Se_BGI | 81.0 [68.7;92.0] | 80.3 [65.3;92.9] | 80.2 [66.1;92.0] | 81.0 [68.4;92.0] | 80.6 [67.7;91.7] | 78.9 [66.4;90.1] | 79.3 [66.4;90.7] |
| Sp_TIB | 99.1 [97.3;1] | 98.9 [96.8;1] | 98.8 [96.8;1] | 98.9 [97.0;1] | 99.3 [97.6;1] | 99.3 [97.7;1] | 99.1 [97.4;1] |
| Sp_Da An | 96.1 [93.0;99.1] | 95.8 [92.3;99.0] | 96.3 [92.9;99.5] | 96.3 [92.9;99.3] | 95.9 [92.5;99.0] | 96.5 [93.2;99.5] | 96.6 [93.5;99.7] |
| SP_BGI | 91.6 [87.5;95.2] | 92.0 [87.4;96.8] | 91.2 [86.9;95.1] | 91.7 [87.4;95.6] | 92 [87.8;96.0] | 91.3 [87.3;95.1] | 91.5 [87.4;95.3] |
| Cov | - | 0.008 [-0.023; 0.056] | 0.006 [-0.024; 0.045] | -0.006 [-0.035; 0.024] | 0.002 [-0.001; 0.008] | 0.002 [-0.001; 0.008] | 0.001 [-0.003; 0.007] |
| Prev 1 | 3.6 [0.6;8.3] | 3.7 [0.4;8.6] | 3.7 [0.5;8.3] | 3.7 [0.5;8.2] | 4.3 [0.7;9.4] | 4.3 [0.8;9.2] | 4.2 [0.8;8.9] |
| Prev 2 | 35.4 [21.7;50.7] | 35.8 [20.6;51.4] | 34.9 [21.2;50.1] | 35.1 [21.2;49.9] | 36.3 [22.2;51.7] | 36.4 [22.3;51.0] | 35.9 [21.9;51.3] |
| Prev 3 | 28.7 [20.2;38.1] | 29.2 [20.0;40.5] | 28.6 [20.1;38.0] | 28.9 [20.2;38.2] | 29.5 [21.2;39.5] | 29.3 [20.7;38.5] | 29.5 [20.7;39.2] |
| deviance | 80.16 | 81.32 | 80.74 | 80.73 | 80.36 | 79.99 | 80.68 |
| bpv | - | 0 [0;1] | 0 [0;1] | 0 [0;1] | 0 [0;1] | 0 [0;1] | 0 [0;1] |

Posteriors means and corresponding 95% credible intervals (CrI) of models without any covariance or one covariance between either two sensitivities or two specificities are displayed in Table 6. There was no evidence, based on visual inspection of the histograms and on 95% credible intervals (not including 0), that adding a covariance term led to a better model fit. Convergence was checked by visual inspection of the trace plots and the Gelman-Rubin statistic with all potential scale reduction factors (psrf) being below 1.05. Adding a covariance term between the sensitivities if Tib Molbiol and Da An led to a slight decrease in the sensitivities of all RT-PCRs, whereas adding covariance's between Tib Molbiol and BGI or between Da An and BGI did not substantially alter the posteriors. Adding covariance's between the specificities, mildly affected the sensitivities, but the specificities were not affected considerably.

Based on the obtained posteriors for the sensitivities, specificities and prevalence, positive and negative predictive values were determined for each test and in each population as depicted at Table 7.

## Discussion

High-throughput quality SARS-CoV-2 diagnostic testing is the pivotal factor and has an indispensable role in the control of the current global pandemic [42]. The pandemic is not over. Thus, accurate, reliable and timely laboratory supported testing and management of the

**Table 7. Predictive values of three tests comparing with three different populations for SARS-COV-2 infection in Addis Ababa, Ethiopia, 2021.**

| Population | Predictive value | TIB Mol Biol | Da An | BGI |
|---|---|---|---|---|
| Population 1 | PPV | 0.765 | 0.450 | 0.266 |
| Population 2 | PPV | 0.979 | 0.923 | 0.841 |
| Population 3 | PPV | 0.972 | 0.897 | 0.795 |
| Population 1 | NPV | 0.993 | 0.994 | 0.992 |
| Population 2 | NPV | 0.905 | 0.915 | 0.898 |
| Population 3 | NPV | 0.929 | 0.936 | 0.923 |

*PPV-positive predictive value, NPV- negative Predictive value.

pandemic is very much important to curve the infection through swiftly to test and care for patients and to trace their contacts for brining to care. However, this efforts might be hampered by a lack or limited information on the test kit quality diagnostic performance by which have great impact on the testing and containment of the virus [9,10,17].

Accordingly, the current study aimed to evaluate three commercially available one step RT-PCR assays in Ethiopia with an overall diagnostic sensitivity of 0.813(0.658–0.938), 0.836 (0.712–0.940) & 0.810(0.687–0.920), and a specificity of 0.991(0.973–1.000), 0.961(0.930–0.991) & 0.916(0.875–0.952) of TIB MOLBIOL, Da An and BGI respectively, which is slightly lower performance than a study done by public health England using BGI test kits of sensitivity 97.4% (86.5 to 99.9%; 95% CI and by Corman et al., as sensitivity of 97.9% [92.8–99.7 by TIB MOLBIOL, Berlin, Germany using E-gene [42,44] and its superior findings than SARS-COV-2 R-GENE (BioMérieux) with 60.2% [49.8–70.0] to 66.3% [56.1–75.6]) and combined with those for N or the RdRp genes (71.4% [61.4–80.1] and 69.4% [59.3–78.3], respectively) reported by Alcoba et al. and it's concordance finding with in positive case of BGI of SARS-CoV-2 in Australia showed high diagnostic power in detecting SARS-CoV-2 with a positive percent agreement of 88.89% (95% CI: 83.4%, 94.3%). the main difference might be study time and sampling of the presumptive cases too. More over all of these studies try to present their finding against the classical "gold standard" approach. However, we used an innovative approach with the Bayesian latent class model (BLCM), which is recently recommended to see the diagnostic tests accuracies in the absence of a perfect gold standard [43–45].

The current results showed a moderate to substantial level of agreement of the inter test agreement beyond chance of the three assays with an average Cohen's kappa value of 0 .55 to 0.72. Which is an inter performance agreement a measure of 0.72(95% CI(0.61–0.82)) among TIB and DaAn assays, 0.61 (95% CI(0.49-.0.72)) b/n TIB and BGI with substantial agreement where as a moderate agreement were reported among DaAn and BGI assays with a kappa value of 0.55(95% CI (0.44–0.67)) via 279 COVID-19 suspected individuals, which was consistent with similar studies(27,51,52) and lower finding compared with Altamimi, Asmaa M., et. al., the difference might be the defined study population [46].

From the current study, we learnt that, TIB and Da An assays correctly identified all negative samples while two assays each failed to correctly identify one different negative sample. GI has been presented in low specificity. The consistent detection of positive samples at different Ct values gives an indication of when to repeat testing and/or establish more stringent in-house cut-off value. The varied performance of different diagnostic assays, mostly with emergency use approvals, for a novel virus is expected. This comparative assay' performance findings may guide country laboratories to determine both their repeat testing Ct range and/or cut-off value [43,47].

Notably, all professional and pharmaceutical, diagnostic developer should take their maximum capacities to get the best quality diagnostic assay to flatten the curve of COVID-19. Likewise, RT-PCR is the current established gold standard approaches for the diagnosis of COVID-19, but it has its own limitation in high false negativity rate. This is more or less concordance with our study findings. Thus, in light of this, further action should be considered to get the most sensitive and specific test kits, that can minimize the false positive and false negative cases in to insignificant level [43,47,48].

We noted that the positivity proportion of these three assays were higher among the age group of 21-30years of the presumptive, and there is no gender specific difference among the participants, which is similar and concordance with prior studies [25,33]. Likewise, the evaluated assays sensitivity diagnostic performances have been improved its levels of sensitivity improved to those of all other kits when one primer-probe combined with other targeted genes, but it takes more time to repeat for confirming the screening test with more sensitive.

Thus, one touch step assays of combining different targets should be considered to have more competitive with cost effectiveness and benefit from combining the results of more than one primer probe set [22,27,31].

In the current study, we are at most the highest level of care and rightfully controlled procedure from the pre analytical to post analytical, which will be the contributing factors for the quality performance difference. Given that, we used similar analysis, similar testing time, similar personnel, sample collection strategy.Moreover we try to use similar written standard procedure, which is all assay solutions were uniformly affected. Thus, the differences in diagnostic sensitivity among the evaluated test kits were purely related to their different components. This finding will give a highlight which target gen is more sensitive while selecting a gene target, quality control, primers and give a clue for pharmaceuticals how to choose alternative test kits for the diagnosis of COVID-19 and that minimizes misdiagnoses of an active SARS-CoV-2 infection [29,49].

The current global pandemic is in a way of highly genomic variation and changing its primary nature that might have a diagnostic variability performance and RT-PCR assay have not yet documented gold standards for their comparative analysis. Accordingly, we followed the STARD-BLCM guidelines by which specificity robust estimates independent of conditional dependencies, fewer data to estimate sensitivity which leads less robust. Prevalence estimates are in line with having a higher prevalence in contact cases and suspects as of the study populations, the predictive values were found influenced by prevalence's. In case of TIB MOLBIOL; there were 2 discordant cases of the primary screening E gene based targeted test compared with a confirmatory of RdRp and also its time taking and resource intensive while repeating those discordant results. On top of this, there might be more false negative clients with Da An and TIB MOLBIOL comparing to BGI; which might be a source of infection in the community as false negative diagnosis [43,44,47].

We report a higher ct value in case of DaAn (31.95±5.6 (in case of N gen & 34.20±5.6 in case of ORF1) and lower ct value in BGI with a mean (± SD) Ct value of 28.78±5.7 by ORF1 of BGI among the positive cases. Higher ct value mean it needs more resources and time, while smaller ct value mean, positive result with in shorter turnaround time(TAT) and cycle of PCR reaction that can be more efficient than the others. In case of DaAn, in case of inconclusive test results and situations like ct value of the test sample is NMT 40 only in one channel b/n FAM and VIC, and no amplification curve in the other channel, the manufacturer recommend to repeat the test.if the result of the retest consist with the original, it can be determined as positive for SARS-CoV-2; if the result of the retest is negative, it can be determined as negative. This leads more strenuous and more tiring among the professional working in this lab that might be pushing looking for and searching more recommend user friendly and single touch/ step test kits in the era of highly demanding laboratory results with in short around time [20,32,33,36].

Among the evaluated assays the laboratory try to perform the SARS-CoV-2 diagnosis based on the availability of the test kits. Among these TIB MOLBIOL urge the test to run two times as a manner of screening by using Egen and confirmatory based RdRp, the final verdict of the test judge based on the RdRp findings and in case of DaAn the final diagnostic result should be based on the agreement of Ngen and ORF. While in case of BGI the final decision of the presumptive to have positive or negative based on only and only one test using ORF based. From these we understand that the former two tests need more time and resources to finalize the result with much lever intensive and not user friendly, which makes significant difference and BGI be more upper weigh and be superior, that concur with different similar studies [35,36,46,47].

In the other hand, we get a chance of feedback about the supply chain management system from the professionals and try to test out the reagent supply in light of sustainable laboratory reagents and consumables have been a pillar in the global response to the COVID-19 pandemic. In this regard, we learnt that the testing laboratories were suffering a lot with erratic supply chain management system.However, this should be expatiated sustainable supplies will go on with to be a key tool for controlling COVID-19 and regress effort by supply chain managers, policy makers and laboratory professionals [31,43,46].

## Limitation of the study

We noted that possibly not enough data to estimate conditional dependencies between sensitivities which is minor differences in posteriors. Due to high estimates for specificities, conditional dependencies between specificities are implausible. Thus, readers should take in to consideration while inferring our findings.

## Conclusion and recommendation

Based on the current research findings, we wind up as Tib Mol Biol performs best, followed by DaAn: the sensitivity of all tests are rather similar, but there are differences with regard to specificity. So for issues and costs associated with false positive, one should go for TIb Mol Biol.; and we understand that these assays are helpful tools for the routine diagnosis of COVID-19 in resource limited settings and could be implemented accordingly considering their prioritization contexts.

Despite the incredible efforts have been made to control the pandemic a global crisis has been aggressively going on. To contain this calamity, based on the recent study finding, the following points should be considered. Large scale study should be conducted and there might be the genetic mutation of the circulating virus during our sampling that might be the source of the performance variation, possibly false negative. Thus, the research team highly recommends the whole genome sequencing to know the circulating variant/strain and to come up a conclusive justification.

## Supporting information

**S1 File. Supplementary material used for the BLCM code for analysis, Addis Ababa, Ethiopia, 2021.**
(R)

**S1 Annex.**
(DOCX)

## Acknowledgments

We are very much gratitude and acknowledge the Addis Ababa University, EPHI, and Yekatit 12 Hospital medical college and Mr. Zerihun Woldesenbet for their strong support and assistance in accessing diverse resources used in the study. Our indisputable gratitude goes to Professor Paul Torgerson (professor of Veterinary Epidemiology Vetsuisse Faculty, Zurich) for his commendable assistance in grammatical review and for proofreading the document. We are also acknowledge all the data collector and study participants.

## Author Contributions

**Conceptualization:** Abay Sisay, Adugna Abera.

**Data curation:** Abay Sisay, Adugna Abera, Boja Dufera, Tujuba Endrias, Geremew Tasew, Abraham Tesfaye, Sonja Hartnack, Dereje Beyene, Adey Feleke Desta.

**Formal analysis:** Abay Sisay, Adugna Abera, Boja Dufera, Tujuba Endrias, Geremew Tasew, Abraham Tesfaye, Sonja Hartnack, Dereje Beyene, Adey Feleke Desta.

**Funding acquisition:** Abay Sisay, Adugna Abera.

**Investigation:** Abay Sisay, Adugna Abera, Boja Dufera, Tujuba Endrias, Geremew Tasew, Abraham Tesfaye, Sonja Hartnack, Dereje Beyene, Adey Feleke Desta.

**Methodology:** Abay Sisay, Adugna Abera, Boja Dufera, Tujuba Endrias, Geremew Tasew, Abraham Tesfaye, Sonja Hartnack, Dereje Beyene, Adey Feleke Desta.

**Project administration:** Abay Sisay, Adugna Abera, Boja Dufera, Tujuba Endrias.

**Resources:** Abay Sisay, Adugna Abera.

**Software:** Abay Sisay, Adugna Abera, Sonja Hartnack, Adey Feleke Desta.

**Supervision:** Abay Sisay, Adugna Abera, Boja Dufera, Tujuba Endrias, Geremew Tasew, Abraham Tesfaye, Sonja Hartnack, Dereje Beyene, Adey Feleke Desta.

**Validation:** Abay Sisay, Adugna Abera, Boja Dufera, Tujuba Endrias, Geremew Tasew, Abraham Tesfaye, Sonja Hartnack, Dereje Beyene, Adey Feleke Desta.

**Visualization:** Abay Sisay, Adugna Abera, Tujuba Endrias, Geremew Tasew, Sonja Hartnack, Dereje Beyene, Adey Feleke Desta.

**Writing – original draft:** Abay Sisay, Adugna Abera, Adey Feleke Desta.

**Writing – review & editing:** Abay Sisay, Adugna Abera, Boja Dufera, Tujuba Endrias, Geremew Tasew, Abraham Tesfaye, Sonja Hartnack, Dereje Beyene, Adey Feleke Desta.

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
