## [Decision Letter · Decision Letter 0]

26 Nov 2021

PONE-D-21-35232Diagnostic accuracy of three commercially available one step RT-PCR assays for the detection of SARS-CoV-2 in resource limited settingsPLOS ONE

Dear Mr Abay Sisay,

Thank you for submitting your manuscript to PLOS ONE. After careful consideration, we feel that it has merit but does not fully meet PLOS ONE’s publication criteria as it currently stands. Therefore, we invite you to submit a revised version of the manuscript that addresses the points raised during the review process.

We look forward to receiving your revised manuscript.

Kind regards,

Basant Giri, Ph.D.

Academic Editor

PLOS ONE

Journal Requirements:

(This work was partially financed by Addis Ababa University through adaptive research and problem solving project, Ref #-PR/5.15/590/12/20. However, the University had not any role in the research methodology design, data collection, analysis and write up of the manuscript. The contents are purely the responsibilities of the authors and did not represent and reflect the view of the funder.)

6.  We note you have included a table to which you do not refer in the text of your manuscript. Please ensure that you refer to Table 7 in your text; if accepted, production will need this reference to link the reader to the Table.

8. Upon re-submitting your revised manuscript, please upload your Supplementary file as Supporting Information files or to a stable, public repository and include the relevant URLs, DOIs, or accession numbers within your revised cover study’s minimal underlying data set as letter. For a list of acceptable repositories, please see http://journals.plos.org/plosone/s/data-availability#loc-recommended-repositories. Any potentially identifying patient information must be fully anonymized.

Reviewers' comments:

Reviewer's Responses to Questions

**Comments to the Author**

1. Is the manuscript technically sound, and do the data support the conclusions?

Reviewer #1: Yes

Reviewer #2: Partly

2. Has the statistical analysis been performed appropriately and rigorously? 

Reviewer #1: Yes

Reviewer #2: Yes

3. Have the authors made all data underlying the findings in their manuscript fully available?

Reviewer #1: Yes

Reviewer #2: Yes

4. Is the manuscript presented in an intelligible fashion and written in standard English?

Reviewer #1: Yes

Reviewer #2: No

5. Review Comments to the Author

Reviewer #1: This is a decent study that evaluates the diagnostic performance of 3 available RT-PCR assays in resource-limited setting in Ethiopia. Experiments and statistical analyses are performed to a high technical standard and are described well. However, major concern with this research is the sample size – is it sufficient enough to draw conclusions on their performance evaluation?

Below are some other comments that must be addressed before it can be published.

Major comments:

1. There are other assays available in Ethiopia. Why are only three of them selected in this study? There should a separate sub-section under Experiments which should clearly explain the rational behind the selection of those assays.

2. In the data quality assurance section, it is mentioned that the assays were performed by following standard operating procedures and is against specific manufacturing recommendations. What does it mean? Does it mean that, all the assays were performed by standard operating procedure and not the manufacturers protocol? If so, what is the rationale behind doing this? If specific manufacturers protocol is not followed, how can we say that the result is from that particular assay? Is the result now trustworthy?

3. The details about the assays must be mentioned at least in SI in detail. The details of assay components, for example, lot#, expiry date, date of analysis etc. must be given.

It is also mentioned that the quality of assay reagents was checked? How was it done; it needs to be mentioned.

4. From figure 1, the threshold cycle for BGI assay is very less compared to others. Significance test must be done between these three. From this data, can we say that BGI assay is sensitive over others? But the BLCM analysis looks contradicting – BGI being the least sensitive. It is important to discuss figure 1 in the main text.

5. More importantly, the conclusions/recommendation about the usage of assays is missing in this research. The authors, after this research, what do they recommend for the reliable SARS-CoV-2 detection? Rather than mentioning the problems only and suggesting the alternative, it is important to recommend something out of this research, otherwise it is ineffective.

Minor comments:

1. Line 42: efforts made

2. Line 45 and 47: not well documented in resource-limited setting, then why is it performed in clinical setting? These two settings do not really line-up well here.

3. Line 178 and 179: inconsistencies in numbering population

4. Line 268: cut-off

5. Table 3 is difficult to follow.

Reviewer #2: Summary of the research and overall impression:

This study is really admirable for the effective diagnosis and control prospective in this current pandemic situation in the resource constraint setting as we are also engaging in similar COVID-19 diagnosis activities despite bearing countless challenges and biohazard risks.

This manuscript aimed to investigate different diagnostic assays available in Ethiopia’s clinical setting using Bayesian latent class model, as PCR is not considered as gold standard. Although, research gap is explained thoroughly, the scientific hypothesis has not been addressed. Manual RNA extraction was conducted, master-mix preparation, amplification and results were analysed as indicated by the manufacturer. The technical aspects of the paper are adequately conveyed and the BLCM method is appropriate to answer the question.

Specific areas for improvement:

Major issue:

1. Please clarify what the author wants to claim in the fourth paragraph. [Section: Available diagnostic platform for the diagnosis of COVID-19, Reference no.: 9]

2. Perhaps, explain what FIND does to provide more insight, as it is used to evaluate diagnostic tests commercially available. [Section: Available diagnostic platform for the diagnosis of COVID-19, Reference no.: 23, Fifth paragraph]

3. No clear indication on critical, timely and containment strategy explanation. [Section: Available diagnostic platform for the diagnosis of COVID-19, Reference no.: 26, Sixth paragraph]

4. TIB targets E and RdRp genes, Da An targets N and ORF1ab genes, and BGI targets ORF 1ab gene. It has clear indication that these three different assays target different genes, please mention how these can be correlated to compare and predict its specificity. As various publications have already claimed that different target genes have different expression levels and detectable range. [Section: Available diagnostic platform for the diagnosis of COVID-19, Last paragraph]

5. The sentence, “The quality of each reagent was checked before the actual laboratory analysis.”, needs a better explanation. What does the word ‘quality’ indicate? How was the quality checked? [Section: Data Quality Assurance, First paragraph]

6. As BLCM assigns individuals to classes based on their probability of being in classes given the pattern of scores they have on indicator variables. Proper class assignment is not guaranteed. Also, because class assignment is based on probabilities, the exact number or percentage of sample members within each class cannot be determined. Please, briefly explain how this study assigns samples to different classes. [Section: Bayesian latent class model (BLCM)]

7. The discussion is well-written, although it could benefit from including other relevant studies/prior works to support your results. Likewise, few parts of discussion are already stated in the introduction, so it would be better if care be taken for making it concise and readable. Also, if possible please break down the reference and include it in-text. (provide reference at the end of sentence rather than at the end of each paragraph) [Section: Discussion]

8. Conclusion is relatively weak. Please, consider revising it and sufficiently summarizing the methods and results.

9. While the study appears to be sound, the language is unclear, making it difficult to follow. I advise the authors to work with a writing coach or copyeditor to improve the flow and readability of the text.

Minor issues:

1. The authors have acquired ethical approval from their respective boards but haven’t mentioned if they have also acquired the informed (verbal or written) consent from the patients themselves.

2. The manuscript should rigorously undergo for plagiarism check.

6. PLOS authors have the option to publish the peer review history of their article (what does this mean?). If published, this will include your full peer review and any attached files.

Reviewer #1: No

Reviewer #2: No

---

## [Author Response · Author response to Decision Letter 0]

4 Dec 2021

Date: 04 December, 2021

To: PLOS ONE Journal editorial office

Dear Editor: 

We are very much glad to write this response that our research output manuscript entitled “Diagnostic accuracy of three commercially available one step RT-PCR assays for the detection of SARS-CoV-2 in resource limited settings, with a manuscript reference number PONE-D-21-35232” has been possible considered for publication in PLOS ONE. We are pleased to have an opportunity to make our manuscript revised and we have greatly appreciated for the reviewers’ high level comments, and suggestions were very helpful for the overall improvement of the manuscript. In revising the manuscript, we have carefully considered reviewers’ comments, academic editor and suggestions on our revised submission. 

Moreover, this work was financed by Addis Ababa University through adaptive research and problem solving project, Ref #-PR/5.15/590/12/20. However, the funders had no role in study design, data collection and analysis, decision to publish, or preparation of the manuscript. The contents are purely the responsibilities of the authors and did not represent and reflect the view of the founder. There was no additional external funding received for this study.

As instructed, we have attempted to succinctly explain changes made in reaction to all comments and reply to each comment in point-by-point fashion as appended here

Response to Academic Editor Comments/Journal requirements 

1. Please ensure that your manuscript meets PLOS ONE's style requirements

Response-Noted

2. About Ethics and Consents

Response- This research has been accomplished after getting ethical approval from EPHI and AAU (EPHI–IRB-259-2020 and 029/20/Lab (AAU, CHS) respectively). All participants’ identifiers were removed and only codes were used throughout the study to keep confidentiality. 

During data and sample collection the data collectors inform each study participant about the purpose and anticipate benefits of the research project and also informed on their full right to refuse, withdraw or completely reject partly or all of their part in the study. Then, we obtained written informed consent from adult study participants and parents or legal guardians of study participants under the age of 18 years to participate in the study and to use their files and records for the purpose of this study. 

Furthermore, this work was done accordance with the principles of Helsinki declaration.

3. Grant information------

Response- Revised accordingly, as

This work was financed by Addis Ababa University through adaptive research and problem solving project, Ref #-PR/5.15/590/12/20. However, the funders had no role in study design, data collection and analysis, decision to publish, or preparation of the manuscript. The contents are purely the responsibilities of the authors and did not represent and reflect the view of the founder. There was no additional external funding received for this study.

6. about ethics statement – 

Response- revised and done accordingly. Please have a look the revised manuscript.

7. about table 7-

Response- revised and done accordingly. Please have a look the revised manuscript.

8. Supporting Information files-

Response- revised and done accordingly. Please have a look the revised manuscript.

Reviewer #1

XXXX-However, major concern with this research is the sample size – is it sufficient enough to draw conclusions on their performance evaluation?

Response- Sample size is very much important for inferring the result for the general population. Accordingly, we have tried to perform our best. We run 3 plates of 96 reagents with their quality control. And there are many published and recommended of similar studies for this type of experiments, even less than this sample size. Most of the experiments recommended sample size is greater than 100. Please refer the following published works and CLSI protocols, as an example…Clin Chem. 1999 Jun; 45(6 Pt 1):882-94. PMID: 10351998, CLSI guide CLSI EP12-A2. 

1. There are other assays available in Ethiopia. Why are only three of them selected in this study? There should a separate sub-section under Experiments which should clearly explain the rationale behind the selection of those assays.

Response- Of course, there were others, but during our study these were the widely available and used commercial available one step RT-PCR tests in the country , as already explained page 6 line number 155 of the submitted manuscript.

2. In the data quality assurance section, it is mentioned that the assays were performed by following standard operating procedures and is against specific manufacturing recommendations. What does it mean? Does it mean that, all the assays were performed by standard operating procedure and not the manufacturer’s protocol? If so, what is the rationale behind doing this? If specific manufacturer’s protocol is not followed, how can we say that the result is from that particular assay? Is the result now trustworthy?

Response- In the ISO standards working on the quality aspects..” Against the …” mean as per the recommendation protocols, which means comparing and contrasting that particular recommendations. It has a synonyms of compared to. In our cases we used the specific recommended SOPs and manufacture recommendations and protocols, then we have compare and contrast accordingly. 

Just to avoid misleading and misinterpretation we have rephrased it as … according to…. Please have a look the revised manuscript line 270.

3. The details about the assays must be mentioned at least in SI in detail. The details of assay components, for example, lot#, expiry date, date of analysis etc. must be given.

It is also mentioned that the quality of assay reagents was checked? How was it done; it needs to be mentioned.

Response-for this comments, we understand that, It’s already existed at the table 1, if any one of the readers need to understand about each assays, the can cross refer the listed references at the column of the table. I don’t know, if we exostively list those things, the document could be not attractive or the fellow readers. 

4. From figure 1, the threshold cycle for BGI assay is very less compared to others. Significance test must be done between these three. From this data, can we say that BGI assay is sensitive over others? But the BLCM analysis looks contradicting – BGI being the least sensitive. It is important to discuss figure 1 in the main text.

Response-The assays or the three different RT- PCRs have different company production with different targets, and that their CT values were also different and their interpretation is quite different as per their experimental recommendation of respective manufacturer. Thus, they should not be the same. That’s why we run the BLCM.

5. More importantly, the conclusions/recommendation about the usage of assays is missing in this research. The authors, after this research, what do they recommend for the reliable SARS-CoV-2 detection? Rather than mentioning the problems only and suggesting the alternative, it is important to recommend something out of this research, otherwise it is ineffective.

Response- We agree with the reviewer’s comment. The ultimate goal of our analysis is to propose the most sensitive and specific test for the detection of SARS Cov2. In addition, based on a population level, to the results of the BLCM, i.e., having a robust estimation of the performance of the tests under field conditions, can be used to adjust the apparent prevalence by for example the Rogan Gladon approach. Accordingly, we revised our manuscript, and please have a look it, at line 507.

6. Line 42: efforts made

Response- done accordingly 

7. Line 45 and 47: not well documented in resource-limited setting, then why is it performed in clinical setting? These two settings do not really line-up well here.

Response- clinical setting Vs resource. These two settings have not line up. 

Our study was done in clinical settings of resource limited settings of less/developing countries. Thus, resource limited settings means which is more related to the scarcity of economy. Or it mean as resource poor or constrained setting ,whereas clinical settings means a hospital, department, outpatient facility, or clinic, or laboratory, public health institutes … etc which health cares were rendered. 

8. Line 178 and 179: inconsistencies in numbering population

Response: the three study populations were selected conveniently and categorized as where they come until the final proposed sample sizes were full filled.

9. Line 268: cut-off

Response- done accordingly 

10. Table 3 is difficult to follow.

Response-it’s just giving an additional information on the demographic of the study participants considering our readers get in-depth understanding on it.

Reviewer #2

Xxxx”… the scientific hypothesis has not been addressed…”

Response: Our main scientific hypothesis is the assessment of comparing the ct levels and then the demographic data, and finally assessing agreement between the PCRs and obtaining robust estimates for diagnostic sensitivities and specificities in the less developed countries. 

Revised accordingly and please have a look the revised manuscript at line 165-171.

1. Please clarify what the author wants to claim in the fourth paragraph. [Section: Available diagnostic platform for the diagnosis of COVID-19, Reference no.: 9]

Response- We are here for trying to explaining countries specific testing strategies: some countries’ recommend and implement COVID-19 testing, for instance in our country, Ethiopia the recommended test for the diagnosis of COVID-19 is RT-PCR. 

2. Perhaps, explain what FIND does to provide more insight, as it is used to evaluate diagnostic tests commercially available. [Section: Available diagnostic platform for the diagnosis of COVID-19, Reference no.: 23, Fifth paragraph]

Response- In the recent pandemic, COVID-19, FIND has been accomplishing a milestone and commendable activities by independent evaluations of assays for SARS-CoV-2 diagnosis in collaboration with WHO and summarizes data for emergency Use Authorization (EUA) and the recommend specific country has to be perform validation/verification accordingly prior to use it.

Addressed accordingly- and this sentences were included in the revised manuscripts. Please have a look it, line 145.

3. No clear indication on critical, timely and containment strategy explanation. [Section: Available diagnostic platform for the diagnosis of COVID-19, Reference no.: 26, Sixth paragraph]

Response- I think, this information has been already explained in the same section, line 133-138, no need of repetition. 

4. TIB targets E and RdRp genes, Da An targets N and ORF1ab genes, and BGI targets ORF 1ab gene. It has clear indication that these three different assays target different genes, please mention how these can be correlated to compare and predict its specificity. As various publications have already claimed that different target genes have different expression levels and detectable range. [Section: Available diagnostic platform for the diagnosis of COVID-19, Last paragraph]

Response- Sure!, they have different targets and different ranges of detection. However these assays used in Ethiopia and other countries in these states .Thus, this is one of the rational of doing this study to come up for some scientific justification of their performance specifications. Accordingly, we use BLCM as a tiebreaker and a bridge to minimize the gaps among the assays

5. The sentence, “The quality of each reagent was checked before the actual laboratory analysis.”, needs a better explanation. What does the word ‘quality’ indicate? How was the quality checked? [Section: Data Quality Assurance, First paragraph]

Response- The quality of each reagent: lot number, expire date, storage conditions, physical leak proof ness and not breakages, etc were checked before the actual laboratory analysis., and also the positive and negative controls were incorporated in all runs.” This sentences is incorporated in the revised manuscripts for a better explanation for our readers, line 269-274.

6. As BLCM assigns individuals to classes based on their probability of being in classes given the pattern of scores they have on indicator variables. Proper class assignment is not guaranteed. Also, because class assignment is based on probabilities, the exact number or percentage of sample members within each class cannot be determined. Please, briefly explain how this study assigns samples to different classes. [Section: Bayesian latent class model (BLCM)]

Response-: In our analysis we followed the Hui Walter approach, thus we modelled the eight different combinations of dichotomized test results (+++, ++-, +-+, +--, -++, -+-, --+, ---) in the three different populations with a multinomial distribution. Here, we assume that there are two classes (negatives and positives). 

Please have a look at the following to illustrate the principle of BLCM (for simplicity just for two tests, thus with four possible combination of dichotomized test results).

T1+T2+: P1*Se1*Se2+(1-P1)*(1-Sp1)*(1-Sp2)

T1+T2-: P1*Se1*(1-Se2)+(1-P1)*(1-Sp1)*Sp2

T1-T2+: P1*(1-Se1)*Se2+(1-P1)*Sp1*(1-Sp2)

T1-T2-: P1*(1-Se1)*(1-Se2)+(1-P1)*Sp1*Sp2

 Thus, with P1 being the prevalence, Se1 and Se2, the sensitivities of both tests, Sp1 and Sp2, the two specificities and T1+/- and T2+/- the dichotomized test results, we assume that there are two “classes” the true positives and the true negatives. For example, among the samples which tested positive for both tests (T1+T2+) we have the true positives (TP), equal to the product of the prevalence and both sensitivities, and the false positives (FP), equal to the product of (1-P) times (1-Sp test 1) and (1-Sp test2). Here we have five unknowns (P and both sensitivities and specificities) but just four equations (and if the total number of n submitted to two tests is known, the fourth frequency of the two possible test combination becomes known, so we just have three degrees of freedom). In this situation the model is not identifiable. Hui and Walter (Hui SL, Walter SD. Estimating the error rates of diagnostic tests. Biometrics. 1980 Mar;36(1):167-71. PMID: 7370371) proposed a solution, by adding a second population from which samples are also analysed by two tests, assuming that the two tests have the same sensitivities and specificities in both populations. This then would add one additional unknown parameter (the prevalence of the second population (so in total six unknown parameters), but also three additional degrees of freedom. So in total the model becomes identifiable. Hui and Walter have shown that if the following condition for S (the number of populations), and T (the number of tests is fulfilled), the model is identifiable

In our case we have three tests and three populations. The aim is to obtain reliable estimates of se and sp, the aim is not to determine class membership of each individual (being truly positive or negative).

This type of Bayesian latent class analysis (also called no gold standard models) is different from the latent class approach where - based on specific patterns of the data or the underlying latent structure) individuals are assigned to classes and class membership could then be used as a variable for further analysis. In this context one would looks with specific metrics if the data are best describes by 1 to n classes). 

In our approach of BLCM we have performed a sensitivity analysis with different priors to assess the importance of the prior information., 

7. The discussion is well-written, although it could benefit from including other relevant studies/prior works to support your results. Likewise, few parts of discussion are already stated in the introduction, so it would be better if care be taken for making it concise and readable. Also, if possible please break down the reference and include it in-text. (provide reference at the end of sentence rather than at the end of each paragraph) [Section: Discussion].

Response-Done and revised accordingly. Please have a look the revised manuscript.

8. Conclusion is relatively weak. Please, consider revising it and sufficiently summarizing the methods and results.

Response- Addressed and revised accordingly

9. While the study appears to be sound, the language is unclear, making it difficult to follow. I advise the authors to work with a writing coach or copyeditor to improve the flow and readability of the text.

Response-To be honest, we tried our best, in this regard, and we did grammatical review in the revised manuscripts. The comment is accepted positively and revised accordingly, it’s an opportunity for the suitability and readability of our manuscript. Please have a look the revised manuscript.

Minor issues:

1. The authors have acquired ethical approval from their respective boards but haven’t mentioned if they have also acquired the informed (verbal or written) consent from the patients themselves.

Response-We did this work after getting the approval of institutional ethics board and we have getting written consent and assent of the study participants….It’s already addressed in the editor comments section.

2. The manuscript should rigorously undergo for plagiarism check.

Response- Thank you very much for your detailed and rigorous review of the manuscript, it helps for the improvement of the manuscript. And we did and write our manuscript using our own words. Hope the revised manuscript get better improvement accordingly.

Looking forward to hearing from you and thank you very much again for your consideration! 

Sincerely, 

Abay

---

## [Decision Letter · Decision Letter 1]

19 Dec 2021

Diagnostic accuracy of three commercially available one step RT-PCR assays for the detection of SARS-CoV-2 in resource limited settings

PONE-D-21-35232R1

Dear Dr. Sisay,

We’re pleased to inform you that your manuscript has been judged scientifically suitable for publication and will be formally accepted for publication once it meets all outstanding technical requirements.

Kind regards,

Basant Giri, Ph.D.

Academic Editor

PLOS ONE

Additional Editor Comments (optional):

Reviewers' comments:

Reviewer's Responses to Questions

**Comments to the Author**

1. If the authors have adequately addressed your comments raised in a previous round of review and you feel that this manuscript is now acceptable for publication, you may indicate that here to bypass the “Comments to the Author” section, enter your conflict of interest statement in the “Confidential to Editor” section, and submit your "Accept" recommendation.

Reviewer #1: All comments have been addressed

Reviewer #2: All comments have been addressed

2. Is the manuscript technically sound, and do the data support the conclusions?

Reviewer #1: Yes

Reviewer #2: Yes

3. Has the statistical analysis been performed appropriately and rigorously? 

Reviewer #1: Yes

Reviewer #2: Yes

4. Have the authors made all data underlying the findings in their manuscript fully available?

Reviewer #1: Yes

Reviewer #2: Yes

5. Is the manuscript presented in an intelligible fashion and written in standard English?

Reviewer #1: Yes

Reviewer #2: Yes

6. Review Comments to the Author

Reviewer #1: All the comments have been addressed. This article can be considered for publication after minor revisions.

Reviewer #2: Went through all the rebuttals provided by authors in the revised manuscript, I found that the authors have given justifiable reasons and explanations for each major and minor issues. Moreover, they have made necessary revisions to the manuscript and are considerable.

Though, the conclusion is still comparatively weaker compared to the respective result/discussion sections and hence recommend to make it scientifically strong. Also, provide the take home message and contributions in the frontiers of molecular diagnostics.

7. PLOS authors have the option to publish the peer review history of their article (what does this mean?). If published, this will include your full peer review and any attached files.

Reviewer #1: No

Reviewer #2: No

---

## [Editor Report · Acceptance letter]

23 Dec 2021

PONE-D-21-35232R1 

Diagnostic accuracy of three commercially available one step RT-PCR assays for the detection of SARS-CoV-2 in resource limited settings 

Dear Dr. Sisay:

I'm pleased to inform you that your manuscript has been deemed suitable for publication in PLOS ONE. Congratulations! Your manuscript is now with our production department. 

Kind regards, 

on behalf of

Dr. Basant Giri 

Academic Editor

PLOS ONE